# The Association of Healthy Aging with Multimorbidity: IKARIA Study

**DOI:** 10.3390/nu13041386

**Published:** 2021-04-20

**Authors:** Alexandra Foscolou, Christina Chrysohoou, Kyriakos Dimitriadis, Konstantina Masoura, Georgia Vogiatzi, Viktor Gkotzamanis, George Lazaros, Costas Tsioufis, Christodoulos Stefanadis

**Affiliations:** 1First Cardiology Clinic, Hippokration Hospital, School of Medicine, National and Kapodistrian University of Athens, 11527 Athens, Greece; alexandra.foscolou@gmail.com (A.F.); dimitriadiskyr@yahoo.gr (K.D.); kmasoura@gmail.com (K.M.); gvogiatz@yahoo.gr (G.V.); glaz35@hotmail.com (G.L.); ktsioufis@gmail.com (C.T.); stefanadis.christodoulos@gmail.com (C.S.); 2Department of Nutrition and Dietetics, School of Health Sciences and Education, Harokopio University of Athens, 17676 Athens, Greece; vik.gkot@gmail.com

**Keywords:** aging, healthy aging, multimorbidity, chronic disease, Ikaria, longevity, Greece

## Abstract

The aim of this study was to evaluate several sociodemographic, lifestyle, and clinical characteristics of the IKARIA study participants and to find healthy aging trajectories of multimorbidity of Ikarian islanders. During 2009, 1410 people (aged 30+) from Ikaria Island, Greece, were voluntarily enrolled in the IKARIA study. Multimorbidity was defined as the combination of at least two of the following chronic diseases: hypertension; hypercholesterolemia; diabetes; obesity; cancer; CVD; osteoporosis; thyroid, renal, and chronic obstructive pulmonary disease. A healthy aging index (HAI) ranging from 0 to 100 was constructed using 4 attributes, i.e., depression symptomatology, cognitive function, mobility, and socializing. The prevalence of multimorbidity was 51% among men and 65.5% among women, while the average number of comorbidities was 1.7 ± 1.4 for men and 2.2 ± 1.4 for women. The most prevalent chronic diseases among men with multimorbidity were hypertension, hypercholesterolemia, and obesity while among women they were hypertension, hypercholesterolemia, and thyroid disease. Multimorbidity was correlated with HAI (Spearman’s rho = −0.127, *p* < 0.001) and for every 10-unit increase in HAI, participants had 20% lower odds of being multimorbid. Multimorbidity in relation to HAI revealed a different trend across aging among men and women, coinciding only in the seventh decade of life. Aging is usually accompanied by chronic diseases, but multimorbidity seems to also be common among younger adults. However, healthy aging is a lifelong process that may lead to limited co-morbidities across the lifespan.

## 1. Introduction

The world’s older population is intensely growing [1]. This aging population is mostly a result of two demographic effects: longevity increase and fertility decline. An increment in life span raises the average age of the population by increasing the numbers of surviving older people [2]. Life expectancy has improved dramatically over recent decades by exceeding the age of 75 in approximately 95 countries [3]. In most of the cases, it is not known whether these old populations are living their extra years in good or poor health; it is almost twenty years since the World Health Organization published statistics called the Healthy Life Expectancy (HALE), defined as the average number of years that a person can expect to live in “full health”, excluding the years lived in less than full health due to disease and/or injury [4].

It is a fact that the number of people with or at risk for long-term conditions is also growing rapidly [5]. Multimorbidity, i.e., the coexistence of two or more health conditions in an individual, is a growing public health challenge, being associated with poorer outcomes and increased use of health and social care services [6]. Even though the prevalence of multimorbidity increases with age, it is not uncommon amongst working-age populations. Actually, it is estimated that multimorbidity affects up to 95% of the primary care population aged 65 years and older [7]. Multimorbidity could consequently affect an individual’s quality of life, well-being, and ability to properly function [8], and despite the accumulation of chronic diseases, aging is not a “disease” but rather a lifelong process that occurs from birth to death. The aging population, less healthy lifestyles, and the increasing incidence of chronic conditions mean that multimorbidity is becoming a plague. This trend is a major health care challenge for every country around the world. Every person should have the opportunity to live a long and healthy life, i.e., to age and remain healthy. WHO defines healthy aging as “the process of developing and maintaining the functional ability that enables well-being in older age” [9]. According to the WHO, in order to be considered as having functional ability a person should be mobile; capable of meeting the basic needs; able to learn, grow, and make decisions; able to build and maintain relationships; and finally able to contribute to the society [10]. 

Taking into account the aforementioned and the lack of current data regarding multimorbidity and healthy aging in Greece, and more specifically in insular Greece, it is necessary to carry out research. Thus, the aims of the present work were (1) to evaluate the sociodemographic, lifestyle, and clinical characteristics of a population whose place of residence (Ikaria island, Greece) has been defined as a place where the environment is conducive to old age and the residents are more likely to reach and exceed the age of 90 [11] compared to other populations and (2) to investigate the association and the age trend between multimorbidity and healthy aging among these islanders.

## 2. Materials and Methods

### 2.1. Sample

The IKARIA study is a prospective, observational cohort investigation initiated in 2009. The aims of the study were to evaluate various baseline biological, clinical, behavioral, and lifestyle characteristics of the adult population (aged 30+) of Ikaria island. A volunteer-based, multistage sampling method was used to enroll 667 men (67 ± 14 years) and 743 women (66 ± 14 years) from all over the island (84% participation rate). The participants for the purpose of this work were divided in multimorbidity groups, i.e., “No multimorbidity” group (*n* = 579) and “Multimorbidity” group (*n* = 828), and/or age groups, i.e., 30–39 (*n* = 15), 40–49 (*n* = 179), 50–59 (*n* = 275), 60–69 (*n* = 317), 70–79 (*n* = 360), 80–89 (*n* = 195), and finally 90+ (*n* = 66). Individuals residing in assisted-living centers were not included in the survey. A group of health scientists (nurses and physicians) with experience in field investigation collected all the required information using a structured, quantitative questionnaire and standard procedures.

### 2.2. Bioethics

The study was approved by the Medical Research Ethics Committee of the First Cardiology Clinic of the University of Athens at Hippokration General Hospital and was carried out in accordance with the Declaration of Helsinki (1989) of the World Medical Association (Scientific Committee and Directory Board of Hippokration Hospital decision 10/29-06-2009). All participants were informed of the study aims and procedures and they provided written informed consent for study participation prior to enrollment.

### 2.3. Measurements

#### 2.3.1. Sociodemographic Characteristics

The sociodemographic characteristics studied included age (years), sex (man/woman), educational status (years of school), professional status (retired/housekeeping/employee/businessman/unemployed), and financial status (mean annual income during the past three years was divided into four categories: low (inability of earnings to cover vital needs), moderate (EUR 6000–9600 per year), good (EUR 9601–18,000 per year), and very good (>EUR 18,000 per year).

#### 2.3.2. Lifestyle Characteristics

Current smokers were defined as those who smoked at the time of interview; former smokers were defined as those who had stopped smoking for at least one year. Non-smokers were defined as those who had never smoked. Physical activity was evaluated in metabolic equivalent (MET) minutes per week using the shortened, translated, and validated-in-Greek version of the self-reported International Physical Activity Questionnaire (IPAQ) [12]. Participants who did not report any physical activity were defined as sedentary. Data on hours of sleep per day were also recorded.

#### 2.3.3. Anthropometric and Clinical Characteristics

Weight and height were measured using standard procedures to attain the volunteer’s body mass index (BMI) (kg/m^2^). Obesity was defined as BMI > 29.9 kg/m^2^. Resting arterial blood pressure was measured three times in the right arm at the end of the physical examination with the subject in the sitting position. People who had blood pressure levels ≥140/90 mmHg or used antihypertensive medications were classified as hypertensive. Fasting blood samples were collected from 08:00 to 10:00 h and after a 12 h overnight fast. All serum samples for biochemical evaluations were immediately centrifuged at 3000× *g* for 10 min at +4 °C. The biochemical evaluation was carried out in the same laboratory according to the criteria of the World Health Organization Reference Laboratories. Blood lipids (i.e., total serum cholesterol, HDL cholesterol, and triglycerides) were measured using a chromatometric enzymatic method on an automatic chemistry analyzer. Hypercholesterolemia was defined as total serum cholesterol levels higher than 200 mg/dL or the use of lipid-lowering agents. Diabetes mellitus type 2 was determined by fasting plasma glucose tests and was defined in accordance with the American Diabetes Association diagnostic criteria (fasting blood glucose levels greater than 125 mg/dL (7 mmol/L) or use of special medication indicating the presence of diabetes) [13]. Finally, participants who stated that they had been diagnosed with malignancy tumor or thyroid or chronic obstructive pulmonary disease (COPD) and/or they have taken treatment for neoplasms, antithyroid, or COPD medications were defined as cancer or thyroid or COPD patients, respectively. Renal disease was defined as creatinine clearance < 30 and osteoporosis as spinal or hip bone mineral density ≥ 2.5 standard deviations below the young adult female reference mean (T-score ≤ −2.5). Cardiovascular disease (CVD) was defined as the presence of coronary heart disease, chronic arrythmias, or stroke. Finally, multimorbidity was defined as the combination of at least two of the above-mentioned chronic diseases, i.e., hypertension, hypercholesterolemia, diabetes mellitus type 2, obesity, thyroid disease, cancer, COPD, CVD, osteoporosis, and renal disease. 

#### 2.3.4. Dietary Habits Assessment

Dietary habits were assessed on a semi-quantitative, validated, and reproducible food frequency questionnaire (FFQ) [14]. Specifically, consumption (in times per week or month) of the main 15 food groups and beverages (i.e., meat and its products, poultry, fish and fisheries, milk and other dairy products, fruits, vegetables, greens and salads, legumes, refined and non-refined cereals, as well as coffee, tea, and soft-drinks) was measured on a weekly or monthly consumption basis. Total energy intake was calculated based on the participants’ responses on the food frequency questionnaire. Furthermore, overall assessment of dietary habits was evaluated through a special diet score (MedDietScore, range 0–55), which assesses adherence to the Mediterranean dietary pattern [15]. Higher values on the score indicate greater adherence to this pattern and, consequently, healthier dietary habits. 

#### 2.3.5. Healthy Aging Index

Since there are multiple definitions for healthy/successful aging, we followed the WHO approach that defines healthy aging as “the process of developing and maintaining the functional ability that enables wellbeing in older age” and we created a Healthy Aging Index (HAI), ranging from 0 to 100. For this, four components were used: depression symptomatology, cognitive function, mobility, and socializing. Depressive symptomatology in the middle-aged individuals was assessed using a translated and validated version of the Zung Self-Rating Depression Scale (ZDRS) (range 20–80), while in individuals above the age of 65, the Geriatric Depression Scale (GDS) (range 0–15) was used [16]. Afterwards, these variables were combined on the basis of z-score—a statistical measurement of a score’s relationship to the mean in a group of scores. Cognitive function was evaluated using the Mini-Mental State Examination (MMSE) translated into the Greek language; it consists of a list of questions, allowing a maximum score of 30 [17]. Mobility was assessed by the time (in minutes) each participant stated they spent walking daily, and socializing was assessed by the frequency of social events on a weekly basis. To better understand the association between multimorbidity and HAI, HAI was further categorized as “Low” if HAΙ did not exceed 69.67, “Moderate” when HAI was between 69.67 and 80.96, and “High” when HAI was at least 80.96.

### 2.4. Statistical Analysis

Continuous variables are presented as mean ± standard deviation (SD) and categorical variables as frequencies. Associations between categorical variables and multimorbidity status were evaluated using chi-squared test, while associations between continuous variables and multimorbidity status were evaluated using Student’s *t*-test. Logistic regression models were used to evaluate the association between healthy aging status (independent factor) as a continuous variable (0–100), as tertiles (Moderate vs. Low or High vs. Low), and multimorbidity (outcome). Stepwise regression analysis was used in order to identify a useful subject of the predictors, since it is a step-by-step iterative construction of a regression model that involves adding or removing potential explanatory variables in succession and testing for statistical significance after each iteration [18]. Finally, logistic regression was also used to find the age trends of healthy aging as High vs. Low in association with multimorbidity. STATA (M. Psarros & Associates, Sparti, Greece) software version 15 was used for all calculations.

## 3. Results

Overall, the prevalence of multimorbidity was 51% among men and 65.5% among women (*p* < 0.001), while the average number of comorbidities was 1.7 ± 1.4 for men and 2.2 ± 1.4 for women (*p* < 0.001). In Figure 1, the prevalence of multimorbidity by sex and age group is summarized. Multimorbidity prevalence showed an increased trend until the age of 90, but in the oldest old, i.e., over 90 years of age, this prevalence presented a decrease both in men (Figure 1A) and women (Figure 1B) similar to that of participants of 60–69 years of age for both men and women. At the same time, hypertension (27%), hypercholesterolemia (23%), and obesity (18%) were found to be the three most dominant morbidities for men (Figure 1C) and hypertension (22%), hypercholesterolemia (18%), and thyroid disease (17%) were most dominant for women (Figure 1D).

In Table 1, basic sociodemographic, lifestyle, dietary, and clinical characteristics of the IKARIA study participants, categorized by multimorbidity status, are depicted. Participants who had simultaneously less than two chronic diseases were more likely to be younger (*p* < 0.001), men (*p* < 0.001), more educated (*p* < 0.001), and current smokers (*p* < 0.001) compared to their counterparts with more than two chronic diseases. Moreover, IKARIA study participants without multimorbidity were more likely to have lower levels of adherence to the Mediterranean diet (*p* < 0.001) and depression symptomatology (*p* = 0.001) but higher levels of HAI (*p* < 0.001) compared to participants with multimorbidity. 

A significant correlation was observed between multimorbidity and HAI score (Spearman rho = −0.127, *p* < 0.001); thus, further analysis, i.e., stepwise logistic regression models, were applied to assess the role of some sociodemographic and lifestyle characteristics in the association between HAI and multimorbidity and find potential mediators of this association (Table 2). In the crude model (Model 1), it was revealed that for every 10-unit increase in HAI, participants had 20% lower odds of having multimorbidity (*p* < 0.001). Following adjustments for age, smoking habits, physical activity level, professional status, years of education, financial status, and hours of sleep (Models 1–8) this association remained the same, but when sex (Model 9) and level of adherence to the Mediterranean diet (Model 10) were taken into account, the above mentioned association was lost (*p* values > 0.05), meaning that sex and MedDietScore could be considered as potential mediators of the HAI and multimorbidity association. 

In Figure 2, effect size measures, i.e., OR, from logistic regression models are illustrated to evaluate the age trends of healthy aging (“High vs. Low” HAI level) in association with multimorbidity. It was revealed that for women with High vs. Low HAI, older age was associated with higher odds of having multimorbidity (R^2^ = 0.087) whereas for men the association was inversed, meaning that as the years go by, the odds of having multimorbidity were decreased (R^2^ = 0.279). However, in both men and women the odds of being multimorbid in the seventh decade of their life were similar.

## 4. Discussion

The purpose of the present work was to explore the prevalence of multimorbidity among male and female participants in Ikaria in the IKARIA study across their lifespan, to evaluate several sociodemographic, lifestyle, and clinical characteristics based on their morbidity status, and to find the age trends of healthy aging in association with multimorbidity. Our results revealed that the prevalence of multimorbidity rises rapidly with increasing age until the age of 90 but the prevalence of multimorbidity in older ages decreases, while hypertension and hypercholesterolemia were found to be the dominant morbidities for both men and women. Another striking finding was that in “the island where people forget to die” people have high mean levels of the healthy aging index, while sex and level of adherence to the Mediterranean diet were found to be mediator factors between HAI and multimorbidity. Finally, the age trends of healthy aging in association with multimorbidity revealed a different trend between men and women, coinciding only at the age range from 70 to 79. 

Ikaria, along with five other specific places in the world (i.e., Sardinia in Italy, Okinawa in Japan, Loma Linda in California, and Nicoya peninsula in Costa Rica) are considered to be the places where people live longer, healthier, and happier [19]. Ikarian people have, actually, a rich culture in family values, tradition, and longevity. This can be explained by several factors including environment, lifestyle, and dietary habits, as well as culture [18]. Human life expectancy is influenced by genetics, environment, and lifestyle, while aging is associated with changes in dynamic biological, psychological, physiological, behavioral, environmental, and social processes [20]. Some age-related changes are benign, such as the external phenotype, while others are associated with declines in the functionality of the senses or/and daily activities as well as with increased susceptibility to disease, frailty, and disabilities. Actually, it has been shown that age is an important risk factor for a number of chronic diseases in humans, and in most of the cases, age can bring with it anxiety and depression or the combination of these two [21,22]. However, an important finding of the present work was that older people, over 90 years old, had a lower prevalence of multimorbidity compared to people in their eighties and the proportion was comparable with that of people in their sixties. This finding is in accordance with Darviri et al.’s suggestion showing that aging process is not necessarily accompanied by disease and/or disability [23]. Additionally, the fact that the prevalence of multimorbidity tends to decrease after a certain age has already been suggested and is probably an effect of healthy survivor bias [24]. Actually, the percentage of Greeks who state that they do not suffer from chronic disease (47%) is similar to the EU average (46%) while, in our study, we found that 41% of the participants had less than two chronic diseases. Based on the Hellenic Statistical Authority’s “Health Interview Survey” conducted in 2019, 19.6% (45% men vs. 55% women) of the population aged 15 and over stated that they suffer from hypertension, a percentage reduced by 6.2% compared to what was recorded in 2014 (20.9%). Hypercholesterolemia were reported by 15.8% (46% men vs. 54% women), an increase of 2.6% compared to the rate recorded in 2014 (15.4%). Thyroid disease was reported by 9.6% of the population (18.7% men vs. 81.3% women) and cancer by 1.9%. Finally, obesity was reported by 16.4% while diabetes is reported to affect 8.0% of the population aged 15 and over (48.5% men, 51.5% women), a percentage reduced by 13.0% compared to 2014 (9.2%), while in EU countries the prevalence of diabetes varies from 3.9–7.9%. In other words, the prevalence for the majority of the chronic conditions found in our study is similar to the prevalence of diseases that the entire Greek population seems to have today [25].

However, staying and feeling healthy is important at any age and coping with the changes that age brings could be considered a great challenge, independent of the age, since healthy aging requires continual reinvention as people pass through landmark ages [26]. Our results show that Ikarians also have a high mean healthy aging score independent of multimorbidity status, and in fact when compared to other Greek islanders, this was found to be proportionally very high [27]. Taking into consideration that the healthy aging index, used in this study, was developed based on the functionality and mobility theory, our findings suggest that aging Ikarian people are potentially important for their society, since they remain a resource to their families, communities and economies until a very old age. Actually, a recent study on Ikarians also concluded that older Ikarians maintain their functionality and have very few medical conditions, reinforcing the previous statement [28].

Finding new hobbies, learning to adapt to changes, and staying physically and socially active are crucial points in order to cope with the new challenges [29,30]. It is a fact that environmental and sustainable improvements that began in the previous century have extended life expectancy with significant improvements in food and clean water availability, better housing and living conditions, reduced exposure to infectious diseases, and access to medical care [31].

Morbidity and multimorbidity have become really difficult challenges for the health system and society [6]. Today, the number of people over 60 years of age is approximately one billion, but by 2050 they are estimated to be two billion. However, along with the increased life expectancy, several geriatric conditions coexist. Aging is associated with molecular and cellular damage, leading to mental and physical disability, an increased risk for disease and consequently mortality [26]. Actually, healthy aging is a complex phenomenon and scientists have not yet come to a safe conclusion about the factors that shape healthy aging. In fact, there is evidence suggesting that sex differences also exist in longevity and morbidity; women are considered to be the favored sex regarding longevity but not in morbidity rates [32]. These notable sex-related differences have been related with risk, clinical expression, treatment response, and the course of age-related neurodegenerative disorders [33]. The sex disparities in health and life expectancy of the last century have been reversed, since men present changes in disease importance and behaviors [34]. Our results have now come to reinforce that morbidity differences between men and women are complex and probably they depend on social, dietary, and behavioral aspects.

With the rapid aging of most European populations, policies on healthy aging have become crucial in preventing the burden of disease, disability, and loss of well-being. Ikarians have a less stressful life compared with that of other Greeks and have always had the opportunity to enjoy the goods of the Mediterranean diet. It should be noted that the Mediterranean diet is not just a diet but also a dietary pattern or an outlook on life or a philosophy that stakeholders should take into consideration. This pattern emphasizes moderation and variety, living in harmony with nature, socializing, sharing and enjoying a meal with other people, moderate alcohol consumption (mostly red wine), having an active lifestyle, and having a siesta after meals [35]. Additionally, a lot of studies have shown the beneficial effects of the Mediterranean diet on several diseases, i.e., CVD, diabetes, cancer, and hypertension [36,37]. Taking all of that together, Ikarians seem to have a combination of characteristics beneficial for their health, longevity, and well-being. Moreover, we revealed that Mediterranean diet could be considered as a mediator between multimorbidity and healthy aging, a finding that could be supported by Kyprianidou et al.’s work, in which the researchers showed that there was a direct association between Mediterranean diet and multimorbidity in another Mediterranean population [38].

Significant demographic changes require adaptations to the way societies are structured across all sectors [33]. Adopting a healthy lifestyle and the philosophy of the Ikarian people is crucial for further improving the long-term health of older populations. However, because changes in individual behavior are required, it could be difficult to achieve improvements in policy dimensions, but since “it is never too early and never too late” to change someone’s lifestyle, it is clear that the earlier behaviors change, the more likely it would be for someone to enjoy a healthier, a more functional, and less stressful life.

### Strengths and Limitations

To the best of our knowledge, this is one of the first studies evaluating the association of multimorbidity with healthy aging based on the WHO theory in Ikaria islanders, i.e., Blue Zoners [18]. However, the observational nature of the cross-sectional design does not allow causal associations to be drawn, but it allows for associations to be seen between the examined variables and participants. Moreover, although the MMSE score is widely used to test cognitive function among the elderly, in this work, the MMSE was also used for persons under 65 years of age; however, it has been shown that MMSE can also evaluate cognitive function among younger ages [39]. Finally, the data used in the present work were from 2009 and since then Greek population has suffered several economic crises that could alter several emotional, cognitive, or clinical aspects.

## 5. Conclusions

As life expectancies rise, so too do expectations for a long and healthy life. Understanding the factors affecting health and aging are crucial. As supported by the evidence of the present work, despite the prevalence of multimorbidity, Ikarians have a satisfactory healthy aging level, while hypertension and hypercholesterolemia are the dominant morbidities in both men and women. Finally, sex and level of adherence to the Mediterranean diet were found to be potential mediators between healthy aging and multimorbidity, while age trends of healthy aging associated with multimorbidity were different among men and women. To conclude, aging is usually accompanied by chronic diseases and multimorbidity but multimorbidity seems to be common among younger ages too. Healthy aging is a lifelong process that can lead to limited co-morbidities across the lifespan; thus, stakeholders should focus on personalized, by sex and age, strategies to enhance behaviors that promote a healthy, long, and functional life despite someone’s morbidity status. 

## Figures and Tables

**Figure 1 nutrients-13-01386-f001:**
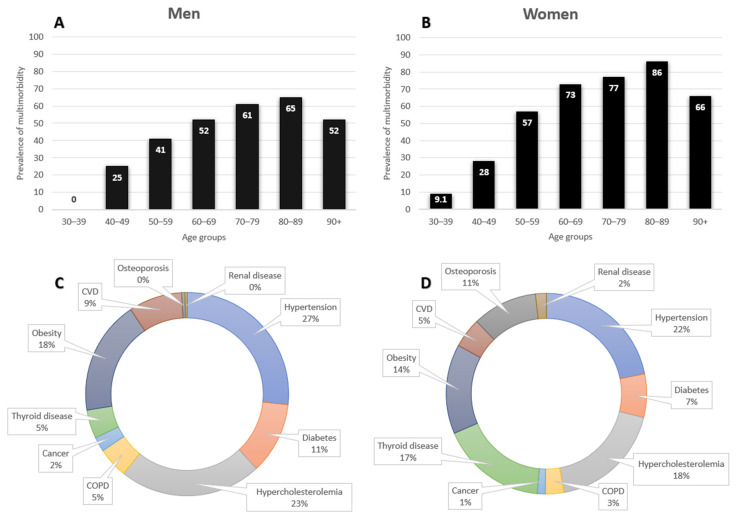
Clustered column charts with the prevalence of multimorbidity across the lifespan among men (**A**) and women (**B**) and pie charts with the proportions of morbidities among men (**C**) and women (**D**).

**Figure 2 nutrients-13-01386-f002:**
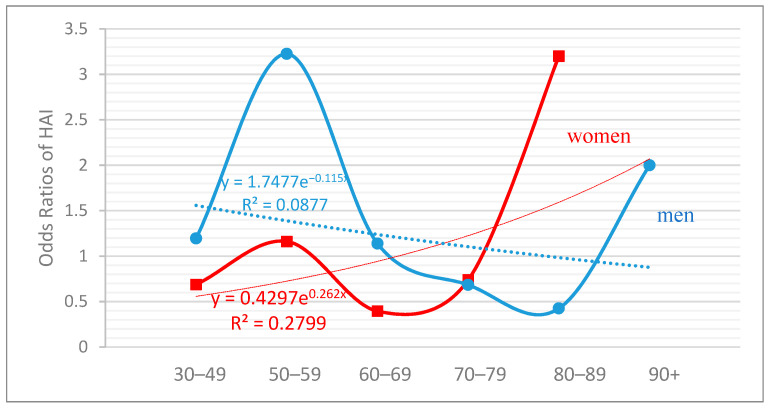
Age trends of healthy aging in association with multimorbidity among men (blue line) and women (red line). Results are presented as odds ratios of the Healthy Aging Index (High vs. Low).

**Table 1 nutrients-13-01386-t001:** Sociodemographic, lifestyle, and clinical characteristics of IKARIA study participants (*n* = 1400) by multimorbidity status (yes/no).

Characteristics	All the Participants (*n* = 1407)	Multimorbidity (No) (*n* = 579)	Multimorbidity (Yes) (*n* = 828)	*p*
**Age (years)**	66 ± 14	66 ± 15	69 ± 12	<0.001
**BMI (kg/m^2^)**	28.5 ± 4.7	26.6 ± 3.7	29.8 ± 4.8	<0.001
**Sex *n* (%)**				
Women	746 (53)	255 (44)	480 (58)	<0.001
Men	661 (47)	324 (56)	348 (42)	
**Years of school**	9.2 ± 3.8	10 ± 3.7	8.0 ± 3.7	<0.001
**Professional status *n* (%)**				
Retired	698 (49.6)	237 (41)	456 (55)	<0.001
Housekeeping	302 (21.5)	98 (16.8)	204 (24.7)	
Employee	224 (15.9)	144 (24.9)	81 (9.8)	
Businessman	178 (12.7)	98 (16.8)	84 (10.1)	
Unemployed	5 (0.4)	2 (0.4)	3 (0.4)	
**Financial status *n* (%)**				
Low	474 (33.7)	162 (28)	298 (36)	0.003
Moderate	620 (44.1)	278 (48)	348 (42)	
Good	272 (19.3)	127 (22)	149 (18)	
Very good	41 (2.9)	12 (2.0)	33 (4.0)	
**Socializing *n* (%)**				
None	134 (9.5)	35 (6.0)	99 (12)	<0.001
<1 per month	507 (36)	197 (34)	307 (37)	
≥1 per month	457 (32.5)	185 (32)	273 (33)	
>1 per week	253 (18)	139 (24)	116 (14)	
Every day	56 (4.0)	23 (4.0)	33 (4.0)	
**Current smokers *n* (%)**	394 (28)	232 (40)	174 (21)	<0.001
**Physical activity status *n* (%) (IPAQ)**				
Low	225 (16)	81 (14)	141 (17)	0.004
Moderate	802 (57)	313 (54)	488 (59)	
High	380 (27)	185 (32)	199 (24)	
**Sleeping hours/day**	7.6 ± 1.7	7.6 ± 1.5	7.6 ± 1.8	0.87
**Total energy intake (kcal/day)**	1599 ± 670	1609 ± 652	1529 ± 680	0.03
**MedDietScore (0–55)**	37 ± 3.2	37 ± 3.2	38 ± 3.2	<0.001
**Depression symptoms (0–60)**	17 ± 11	16 ± 10	18 ± 12	0.001
**MMSE (0–30)**	25 ± 4.7	25 ± 4.9	25 ± 4.6	0.67
**Hypertension *n* (% yes)**	686 (49)	98 (17)	588 (71)	<0.001
**Hypercholesterolemia *n* (% yes)**	565 (40)	93 (16)	472 (57)	<0.001
**Diabetes *n* (% yes)**	250 (18)	18 (3.1)	232 (28)	<0.001
**Obesity *n* (% yes)**	464 (33)	75 (13)	389 (47)	<0.001
**CVD *n* (% yes)**	181 (13)	7 (1.2)	174 (21)	<0.001
**Cancer *n* (% yes)**	70 (4.9)	6 (1.1)	44 (5.3)	<0.001
**Renal disease *n* (% yes)**	29 (2.1)	1 (0.2)	28 (3.4)	<0.001
**Thyroid disease *n* (% yes)**	339 (24)	49 (8.4)	290 (35)	<0.001
**Chronic Obstructive Pulmonary Disease *n* (% yes)**	104 (7.4)	5 (0.9)	99 (12)	<0.001
**Osteoporosis *n*(% yes)**	188 (13)	14 (2.4)	174 (21)	<0.001
**Healthy Aging Score (0–100)**	73 ± 14	76 ± 13	72 ± 14	<0.001

Values are presented as frequencies (%) or mean ± standard deviation. *p*-values derived from Student’s *t*-test for continuous variables or the chi-square test for the categorical variables.

**Table 2 nutrients-13-01386-t002:** Results from stepwise logistic regression models evaluating the association between HAI and multimorbidity (outcome).

	OR	95% CI	*p*
Model 1: HAI (0–100)	0.978	0.970–0.986	<0.001
Model 2: Model 1 + age (years)	0.987	0.979–0.996	0.004
Model 3: Model 2 + smoking status (y/n)	0.984	0.976–0.993	0.001
Model 4: Model 3 + IPAQ	0.981	0.972–0.991	<0.001
Model 5: Model 4 + professional status	0.982	0.972–0.992	<0.001
Model 6: Model 5 + years of education	0.987	0.976–0.997	0.01
Model 7: Model 6 + financial status (low/moderate/good/very good)	0.986	0.975–0.996	0.008
Model 8: Model 7 + hours of sleep	0.987	0.976–0.998	0.02
Model 9: Model 8 + sex (*m*/*w*)	0.990	0.979–1.002	0.09
Model 10: Model 9 + MedDietScore (0–55)	0.995	0.981–1.01	0.54

Results are presented as odds ratios (OR) and 95% confidence interval. y = yes, n = no.

## Data Availability

The data presented in this study are available on request from the corresponding author. The data are not publicly available due to privacy and ethical restrictions.

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
