# Peer review of "The Association of Healthy Aging with Multimorbidity: IKARIA Study"

_nutrients, 2021, doi:10.3390/nu13041386_

Round 1

Reviewer 1 Report

The article studies a topic of interest with a sample known since 2009. Both the design and the parameters studied provide data on the ageing of the elderly and the environment.
I find that the authors do not show the importance and the results of the study in the abstract or in the conclusions, it should be rewritten with more direct information about what was obtained.

Author Response

Comment #1: The article studies a topic of interest with a sample known since 2009. Both the design and the parameters studied provide data on the ageing of the elderly and the environment. I find that the authors do not show the importance and the results of the study in the abstract or in the conclusions, it should be rewritten with more direct information about what was obtained.

Reply #1: We would like to thank the Reviewer very much for the time spent on our work, as well as the useful comments made that helped us, we believe, to improve the presentation of our findings. We have now changed the way our abstract presents our results and conclusions. (pls see the new abstract). Additionally, we have now included a graphical abstract.

Reviewer 2 Report

nutrients-1164308

This manuscript is a cross sectional study from 2009 and tries to evaluate the sociodemographic, lifestyle and clinical characteristics of a population that their place of residence (Ikaria island, Greece) has been defined as a place where the environment is conductive to old age and their residents are more likely to reach and exceed the age of 90 compared to other populations, and, finally, to investigate the association between multimorbidity and healthy aging among these islanders and to explore the age trajectories of healthy aging in relation to multimorbidity.

The document has some aspects that need to be improved. The first is that the data are from 2009, and this should be commented on in limitations.

A Healthy Aging Index (https://academic.oup.com/biomedgerontology/article/69/4/479/547492) was published in 2014 and ranges from 0 to 10. After reading the results I feel that the HAI created by the authors is not the same, and I do not see if it is a scale created by the authors themselves to know its validity or if it is a scale previously used and validated in another field. If it is created by the authors, the metric properties of the HAI should be known and much more information on how this index is calculated would also be needed.

I feel that more justification information should be added in the introduction.

The methods are detailed, but there is a lack of results in relation to IPAQ, anthropometric data, serum markers, dietary patterns, GDS MMSE... The authors have not provided results on these variables and therefore there is no discussion of them.

I have attach further comments on the paper.

Author Response

Comment #1: This manuscript is a cross sectional study from 2009 and tries to evaluate the sociodemographic, lifestyle and clinical characteristics of a population that their place of residence (Ikaria island, Greece) has been defined as a place where the environment is conductive to old age and their residents are more likely to reach and exceed the age of 90 compared to other populations, and, finally, to investigate the association between multimorbidity and healthy aging among these islanders and to explore the age trajectories of healthy aging in relation to multimorbidity. The document has some aspects that need to be improved. The first is that the data are from 2009, and this should be commented on in limitations.

Reply #1: We would like to thank the Reviewer very much for the time spent on our work, as well as the useful comments made that helped us, we believe, to improve the presentation of our findings. Please see the revised manuscript and the new limitation section. It is now added that “the data used in the present work were from 2009 and since then Greek population has suffered several economic crises that could alter several emotional, cognitive or clinical aspects”. (pls see p.9 lines 341-343)

Comment #2: A Healthy Aging Index (https://academic.oup.com/biomedgerontology/article/69/4/479/547492) was published in 2014 and ranges from 0 to 10. After reading the results I feel that the HAI created by the authors is not the same, and I do not see if it is a scale created by the authors themselves to know its validity or if it is a scale previously used and validated in another field. If it is created by the authors, the metric properties of the HAI should be known and much more information on how this index is calculated would also be needed.

Reply #2: Actually, our HAI is not the same as this of Sanders et al. We have followed the WHO approach who defines healthy aging as “the process of developing and maintaining the functional ability that enables wellbeing in older age” and thus we created a Healthy Aging Index (HAI), ranging from 0 to 100. This index was created on the basis of z-scores and it was validated using a widely used index (SAI index) of Tyrovolas et al, 2014, derived from the MEDIS (MEDiterranean Island Study). The correlation between these two indexes was found to be 0.80, i.e., very high.

Tyrovolas S, Haro JM, Mariolis A, Piscopo S, Valacchi G, Tsakountakis N, Zeimbekis A, Tyrovola D, Bountziouka V, Gotsis E, Metallinos G, Tur JA, Matalas AL, Lionis C, Polychronopoulos E, Panagiotakos D. Successful aging, dietary habits and health status of elderly individuals: a k-dimensional approach within the multi-national MEDIS study. Exp Gerontol. 2014 Dec;60:57-63. doi: 10.1016/j.exger.2014.09.010.

Comment #3: I feel that more justification information should be added in the introduction.

Reply #3: We have now added a few more sentences, explaining why the issue addressed in the manuscript deserves attention and why it is of great importance. (pls see p. 2 lines 50-53).

Comment #4: The methods are detailed, but there is a lack of results in relation to IPAQ, anthropometric data, serum markers, dietary patterns, GDS MMSE... The authors have not provided results on these variables and therefore there is no discussion of them.

Reply #4: Thank you very much for this comment. Indeed, several variables are not presented in our results, however all of the above-mentioned variables have been used to define and create other variables like “multimorbidity” or “HAI” and thus we think that they should be shown in the methodology section. In other words, we describe the way we “diagnosed” our participants. Our aim was to present in detail the methods used to create our key variables (i.e., multimorbidity and HAI), without necessarily, as we believe, to present the individual characteristics in the results, since we would deviate from our main goal.

Comment #5: I have attach further comments on the paper.

Reply #5: All the attached comments have now been answered. (pls see the revised manuscript, ref.9,11,22,24, limitation section and table 1)

Reviewer 3 Report

The Authors present an interesting study about the relationship between multimorbidity and “healthy-aging”.

The paper is generally well written, although some minor mistakes are present: a general language revision may improve the readability of the study.

My major concern is related to the use to word “trajectories” both in the title and throughout the manuscript. In my opinion, this word refers to the “pathway” followed by a person or a group of persons in time. However, the study design is cross-sectional: results should be interpreted with caution due to possible effects arising from such design (e.g. cohort effect and selection bias for older persons). This concept further expands to figure 2, which may be misleading.

Other comments:

  • (2.1 – Sample) It would be interesting to know the age-stratified participation rate as well as the absolute number of study participants for each group in order to better describe the study sample.
  • (2.3.3 – Anthropometric and Clinical characteristics) How did the Authors retrieve the diagnoses of neoplasms, COPD, and thyroid diseases? From National registries? Self-reported in questionnaires? Revision of medical charts by physicians? The same question applies to CVDs.
  • The general and/or age-stratified prevalence of each of the chronic conditions reported was similar to those reported in literature for Greece or Southern Europe?
  • (2.3.5 Healthy Aging Index) A better description of the methodology used to build the HAI is necessary, in my opinion.
    • To me, it is not clear how the two tools for depression description were “combined” together obtaining a score ranging between 0 and 60.
    • Furthermore, which version of the GDS was used (15 items? 30 items?) ?
    • Was the MMSE used also for persons under 65 years old?
    • How did the Authors combined all information to obtain the HAI? Were z-score used? Was any weight introduce in the score?
    • Was the score previously validated?
  • (Results): the Authors state that “for every 10-units increase in HAI, participants had 22% lower odds of having multimorbidity”. However, in consideration of the relationship between beta coefficient and odds ratio in logistic regression, this is probably a mistake (OR for HAI-10 points should be 0.80).
  • The Authors state that “…when sex .. and level of adherence to the Mediterranean diet were take into account the … association was lost, meaning that sex and MedDietScore were mediators of the HAI and multimorbidity association”. In consideration of the complexity of mediation analysis, I think the message in this sentence is too strong: the Authors should at least provide references supporting the methodology used (stepwise regression, lack of significant association between exposure and outcome) to identify mediators. Furthermore, the whole causal pathway between multimorbidity and healthy aging is extremely tangled, in particular in older age and the lack of longitudinality in the presented study prevent, in my opinion, any causal inference (included the one on mediators).
  • (discussion): a comparison with the results from this study with the literature is important in the discussion. For example, the fact that the prevalence of multimorbidity tends to decrease after a certain age has been already showed in several study and is probably effect of an healthy survivor bias. The Authors also state that “people have high mean levels of healthy aging index”, however they do not report any comparison for HAI with other population nor normative values.

Author Response

Comment #1: The Authors present an interesting study about the relationship between multimorbidity and “healthy-aging”. The paper is generally well written, although some minor mistakes are present: a general language revision may improve the readability of the study.

Reply #1: We would like to thank the Reviewer very much for the time spent on our work, as well as the useful comments made that helped us, we believe, to improve the presentation of our findings. The manuscript is now language revised. (pls see throughout the manuscript)

Comment #2: My major concern is related to the use to word “trajectories” both in the title and throughout the manuscript. In my opinion, this word refers to the “pathway” followed by a person or a group of persons in time. However, the study design is cross-sectional: results should be interpreted with caution due to possible effects arising from such design (e.g. cohort effect and selection bias for older persons). This concept further expands to figure 2, which may be misleading.

Reply #2: Indeed, a longitudinal study design of the same individuals across time, is the appropriate way of testing trajectories; in our work, we have used the term “trajectories” since we assumed different sampling, but under the same referent population. However, since the Reviewer believes that the use of this term is inappropriate in our case, we have changed this term throughout the text with other phrases being appropriate for the description of relationships in cross-sectional studies (i.e., associations). (pls see throughout the revised manuscript)

Comment #3: (2.1 – Sample) It would be interesting to know the age-stratified participation rate as well as the absolute number of study participants for each group in order to better describe the study sample.

Reply #3: We have included in the methodology section that the overall participation rate was 84% as well as the absolute numbers of study participants for multimorbidity and age groups (pls see p.2 line 73-77 and Table 1).

Comment #4: (2.3.3 – Anthropometric and Clinical characteristics) How did the Authors retrieve the diagnoses of neoplasms, COPD, and thyroid diseases? From National registries? Self-reported in questionnaires? Revision of medical charts by physicians? The same question applies to CVDs.

Reply #4: Thank you for this comment. We have now provided further information regarding the way we retrieved the diagnoses of neoplasms, COPD, and thyroid diseases. “Participants who stated that they have ever been diagnosed with malignancy tumor or thyroid or chronic obstructive pulmonary disease (COPD) and/or they have taken treatment for neoplasms, antithyroid or COPD medications were defined as cancer or thyroid or COPD patients, respectively.” (pls see p.3 lines 121-123)

Comment #5: The general and/or age-stratified prevalence of each of the chronic conditions reported was similar to those reported in literature for Greece or Southern Europe?

Reply #5: Actually, the percentage of Greeks who state that they do not suffer from chronic disease (47%) is similar to the EU average (46%) while, in our study, we found that 41% of the participants had less than two chronic diseases. Based on the Hellenic Statistical Authority’s “Health Interview Survey” conducted in 2019, 19.6% (45% men vs. 55% women) of the population aged 15 and over stated that they suffer from hypertension, a percentage reduced by 6.2% compared to what was recorded in 2014 (20.9%). Hypercholesterolemia were reported by 15.8% (46% men vs. 54% women), an increase of 2.6% compared to the rate recorded in 2014 (15.4%). Thyroid disease was reported by 9.6% of the population (18.7% men vs. 81.3% women) and cancer by 1.9%. Finally, obesity was reported by 16.4% while diabetes is reported to affect 8.0% of the population aged 15 and over (48.5% men, 51.5% women), a percentage reduced by 13.0% compared to 2014 (9.2%). In other words, the prevalence for the majority of the chronic conditions found in our study, is similar to the prevalence of diseases that the entire Greek population seems to have today. (pls see pgs. 7-8, lines 256-274)

Comment #6: (2.3.5 Healthy Aging Index) A better description of the methodology used to build the HAI is necessary, in my opinion. To me, it is not clear how the two tools for depression description were “combined” together obtaining a score ranging between 0 and 60.

Reply #6: Thank you for this comment. Actually, these two tools for depression description (ZDRS and GDS) were “combined” on the basis of standard scores i.e., z-scores. Z-score was used since it is a statistical measurement of a score’s relationship to the mean in a group of scores (pls see p. 4 line 151-154)

Comment #7: Furthermore, which version of the GDS was used (15 items? 30 items?) ?

Reply #7: We have used the GDS score ranging from 0 -15 (15 items). (pls see p. 4 line 151)

Comment #8: Was the MMSE used also for persons under 65 years old?

Reply #8: Yes, the MMSe was used also for persons under 65 years of age. However, although the MMSE index is widely used to test cognitive function among the elderly, it has been shown that it can assess cognitive function among younger ages. This has been now added in the limitation section. (pls see p. 9 lines 338-341)

Nagaratnam, J.M.; Sharmin, S.; Diker, A.; Kwang Lim, W.; Maier, A.B. Trajectories of Mini-Mental State Examination Scores over the Lifespan in General Populations: A Systematic Review and Meta-Regression Analysis. Clin Gerontol. 2020, 1-10.

Comment #9: How did the Authors combined all information to obtain the HAI? Were z-score used? Was any weight introduce in the score?  Was the score previously validated?

Reply #9: This index was created on the basis of z-scores and it was validated using a widely used index (SAI index) of Tyrovolas et al, 2014, derived from the MEDIS Study. The correlation between these two indexes was found to be 0.80, i.e., very high.

Tyrovolas S, Haro JM, Mariolis A, Piscopo S, Valacchi G, Tsakountakis N, Zeimbekis A, Tyrovola D, Bountziouka V, Gotsis E, Metallinos G, Tur JA, Matalas AL, Lionis C, Polychronopoulos E, Panagiotakos D. Successful aging, dietary habits and health status of elderly individuals: a k-dimensional approach within the multi-national MEDIS study. Exp Gerontol. 2014 Dec;60:57-63. doi: 10.1016/j.exger.2014.09.010.

Comment #10: (Results): the Authors state that “for every 10-units increase in HAI, participants had 22% lower odds of having multimorbidity”. However, in consideration of the relationship between beta coefficient and odds ratio in logistic regression, this is probably a mistake (OR for HAI-10 points should be 0.80).

Reply #10: Thank you very much for this comment. Indeed, OR for HAI-10 points is 0.80, i.e., 20% lower odds. This was a typo mistake! It is now corrected. (pls see p. 6 line 207)

Comment #11: The Authors state that “…when sex .. and level of adherence to the Mediterranean diet were take into account the … association was lost, meaning that sex and MedDietScore were mediators of the HAI and multimorbidity association”. In consideration of the complexity of mediation analysis, I think the message in this sentence is too strong: the Authors should at least provide references supporting the methodology used (stepwise regression, lack of significant association between exposure and outcome) to identify mediators. Furthermore, the whole causal pathway between multimorbidity and healthy aging is extremely tangled, in particular in older age and the lack of longitudinality in the presented study prevent, in my opinion, any causal inference (included the one on mediators).

Reply #11: Taken already into account the cross-sectional nature of our study, we added in the limitation section that “the observational nature of the cross-sectional design does not allow causal associations to be drawn, but it allows associations between the examined variables and participants.” (pls see limitation section p.9 lines 336-338). Moreover, we modified the sentence that the Reviewer mentioned so that our statement and our message are not too strong (pls see p.6 lines 196, 211-212) and we added more information regarding the stepwise regression method in the statistical analysis section (pls see p. 4, lines 168-171).

Comment #12: (discussion): a comparison with the results from this study with the literature is important in the discussion. For example, the fact that the prevalence of multimorbidity tends to decrease after a certain age has been already showed in several study and is probably effect of an healthy survivor bias. The Authors also state that “people have high mean levels of healthy aging index”, however they do not report any comparison for HAI with other population nor normative values.

Reply #12: We have now added the Reviewer’s suggestion that the prevalence of multimorbidity tends to decrease after a certain age due to healthy survivor bias effect (pls see p. 7 lines 256-260) Finally, indeed, we haven’t reported any comparisons for HAI with other populations, but this is due to the fact that this index was not used elsewhere. However, since this index was validated using the SAI score from the MEDIS study (pls see reply #9) we have already mentioned that “ Our results have now come to add that Ikarians have also a high mean healthy aging score independently of multimorbidity status, and in fact when compared to other Greek islanders this was found to be proportionally very high [27].” (pls see p.8 lines 278-280)

Round 2

Reviewer 2 Report

The authors have greatly improved the manuscript but I still consider that all the variables described in Methods should be included in the results. It is an analysis criterion in the articles when performing a systematic review, therefore, we know that it is a methodological flaw in the manuscript.
In addition, these variables should be analyzed as confounding variables.

Author Response

Comment #1: The authors have greatly improved the manuscript but I still consider that all the variables described in Methods should be included in the results. It is an analysis criterion in the articles when performing a systematic review, therefore, we know that it is a methodological flaw in the manuscript. In addition, these variables should be analyzed as confounding variables.

Reply #1: We would like to thank the Reviewer very much for the time spent on our work, as well as the useful comments made that helped us, we believe, to improve the presentation of our findings. All variables presented in the Methodology section are now depicted in Table 1 and we have now included these variables (excluding all the variables we’ve used to create HAI or multimorbidity status, to avoid collinearity) in the analyses of Table 2. (pls see the new Table 1 and Table 2)

Reviewer 3 Report

The Authors answered to each point of my previous comments. The revised text presents a clearer picture of the methodology used, allowing the readers to form their ideas about the interesting results presented.

Author Response

Comment #1: The Authors answered to each point of my previous comments. The revised text presents a clearer picture of the methodology used, allowing the readers to form their ideas about the interesting results presented.

Reply #1: We would like to thank the Reviewer very much for the time spent on our work!